# The STRATAA study protocol: a programme to assess the burden of enteric fever in Bangladesh, Malawi and Nepal using prospective population census, passive surveillance, serological studies and healthcare utilisation surveys

Thomas C Darton,[1,2,3] James E Meiring,[2] Susan Tonks,[2] Md Arifuzzaman Khan,[4] Farhana Khanam,[4] Mila Shakya,[5] Deus Thindwa,[6] Stephen Baker,[1,7,8] Buddha Basnyat,[5,7] John D Clemens,[4,9] Gordon Dougan,[8] Christiane Dolecek,[7,10] Sarah J Dunstan,[11] Melita A Gordon,[6,12] Robert S Heyderman,[6,13] Kathryn E Holt,[14,15] Virginia E Pitzer,[16] Firdausi Qadri,[4] K Zaman,[4] Andrew J Pollard,[2]  on behalf of the STRATAA Study Consortium

► Prepublication history and additional material are available. To view these files please visit the journal online (http://dx.doi.org/10.1136/bmjopen-2017-016283)

TCD and JEM contributed equally.

For numbered affiliations see end of article.

**Correspondence to**
Dr James E Meiring; james.meiring@paediatrics.ox.ac.uk

## ABSTRACT

**Introduction** Invasive infections caused by *Salmonella enterica* serovar Typhi and Paratyphi A are estimated to account for 12–27 million febrile illness episodes worldwide annually. Determining the true burden of typhoidal *Salmonellae* infections is hindered by lack of population-based studies and adequate laboratory diagnostics. The Strategic Typhoid alliance across Africa and Asia study takes a systematic approach to measuring the age-stratified burden of clinical and subclinical disease caused by typhoidal *Salmonellae* infections at three high-incidence urban sites in Africa and Asia. We aim to explore the natural history of *Salmonella* transmission in endemic settings, addressing key uncertainties relating to the epidemiology of enteric fever identified through mathematical models, and enabling optimisation of vaccine strategies.

**Methods/design** Using census-defined denominator populations of ≥100 000 individuals at sites in Malawi, Bangladesh and Nepal, the primary outcome is to characterise the burden of enteric fever in these populations over a 24-month period. During passive surveillance, clinical and household data, and laboratory samples will be collected from febrile individuals. In parallel, healthcare utilisation and water, sanitation and hygiene surveys will be performed to characterise healthcare-seeking behaviour and assess potential routes of transmission. The rates of both undiagnosed and subclinical exposure to typhoidal *Salmonellae* (seroincidence), identification of chronic carriage and population seroprevalence of typhoid infection will be assessed through age-stratified serosurveys performed at each site. Secondary attack rates will be estimated among household contacts of acute enteric fever cases and possible chronic carriers.

## Strengths and limitations of this study

► The study is designed with a comprehensive multicomponent epidemiological approach, nesting passive surveillance, serosurveillance and healthcare utilisation surveys within a demographic census population to accurately determine the age-stratified burden of enteric fever.

► The diversity of field sites in Africa and Asia will provide data on typhoid epidemiology and transmission from a range of differing epidemiological settings.

► The combination of a traditional epidemiological approach with novel laboratory methods for the diagnosis of febrile illness and investigation of the host and pathogen genetics and antimicrobial resistance determinants provides a unique platform to study this disease.

► Practical limitations include the sharing and standardisation of clinical definitions, data and sample collection methods and laboratory assays, based on the facilities and staff available and community requirements at each site.

► The number of field sites included in this current protocol is limited to three; all of these sites are densely populated urban settings with likely high incidence of enteric fever transmission. The degree to which our data may be extrapolated to other settings and countries remains to be explored.

**Ethics and dissemination** This protocol has been ethically approved by the Oxford Tropical Research Ethics Committee, the icddr,b Institutional Review Board, the

BMJ

Malawian National Health Sciences Research Committee and College of Medicine Research Ethics Committee and Nepal Health Research Council. The study is being conducted in accordance with the principles of the Declaration of Helsinki and Good Clinical Practice. Informed consent was obtained before study enrolment. Results will be submitted to international peer-reviewed journals and presented at international conferences.
**Trial registration number**  ISRCTN 12131979.
**Ethics references**  Oxford (Oxford Tropical Research Ethics Committee 39-15). Bangladesh (icddr,b Institutional Review Board PR-15119). Malawi (National Health Sciences Research Committee 15/5/1599). Nepal (Nepal Health Research Council 306/2015).

## BACKGROUND

*Salmonella enterica* serovars Typhi (*S* Typhi) and Paratyphi A (*S* Paratyphi A) are human-restricted pathogens transmitted by faeco-oral ingestion. The ensuing disease, enteric fever (or 'typhoid fever'), is a non-specific febrile illness which affects an estimated 12–27 million people worldwide each year, resulting in 129 000–223 000 deaths.[1–3] Despite a dramatic reduction in incidence over the last century in most high-income countries, continuing inadequate access to clean water and increasing intercontinental spread of multiply antibiotic-resistant strains hampers disease control efforts, especially in resource-limited settings.[3–5] The current burden of disease is highest among children and young adults in South and Southeast Asia[1–3] but is increasingly recognised in sub-Saharan Africa.[6]

Recently, a new generation of Vi-conjugate enteric fever vaccines, suitable for use in infants and providing longer-lasting protection than those previously licensed,[7–10] have become available. Determining how and where interventions such as vaccination may be best deployed is difficult due to a lack of population-based incidence studies and inaccurate diagnostic tests.[1–3] Improving disease burden estimates and providing data on the epidemiology and transmission of *S* Typhi and *S* Paratyphi A to inform mathematical models[11–13] could improve the evidence necessary to design comprehensive and effective disease control programmes.[14]

### Burden of disease and diagnostics

Recent meta-analyses examining global causes of morbidity and mortality estimate that a significant burden of disease worldwide, and especially in South and Southeast Asia, may be attributed to enteric fever.[15–17] The margin of error on these estimates is wide, however; much of the uncertainty regarding the burden of disease caused by typhoidal *Salmonella* is due to the unavailability of accurate diagnostics or misclassification of non-specific disease presentations, in addition to a general lack of data.[18] Availability of antibiotics from local pharmacies without prescription, frequent misdiagnosis as malaria, dengue or other febrile illnesses, or the avoidance of hospital due to fear or expense are all likely to result in an underestimate of the true numbers of cases.[19] In contrast, the widespread use of suboptimal diagnostic tests such as the Widal test in areas where exposure to similar bacteria

in the environment occurs may lead to inaccurate overdiagnosis of the condition.[20] Previous studies and systematic reviews have attempted to address this inaccuracy by calculating the imprecision associated with insensitive blood culture methods, and applying this correction to blood culture confirmed case numbers.[1 2]

In addition to a better understanding of the disease burden attributable to typhoidal *Salmonellae*, improved diagnostics and biomarkers are required to improve individual case management.[21] The current gold-standard diagnostic test for typhoid/paratyphoid is blood culture; while this test provides the causative isolate, thereby allowing susceptibility testing and further typing methods to be performed, most laboratories in resource-limited settings lack the infrastructure to perform these assays. Even in ideal conditions, blood culture requires a significant volume of blood to enhance diagnostic yield and is relatively insensitive even in highly controlled human challenge settings where sensitivity reaches 80%.[22] Highly sensitive and specific diagnostic tests are a fundamental requirement for surveillance and vaccine efficacy studies, however, both for measuring the number of cases avoided/prevented, but also for reassuring policymakers and funders that vaccine implementation is worth investment. While several rapid serological diagnostic tests have recently been developed, these still rely on a few selected antigens, which are known to be non-specific in endemic communities.[23 24] Newer diagnostic modalities are in development, however, which have used serum banks from multiple countries and samples collected from controlled human infection models to identify putative novel antigens for serological assay and further high-sensitivity approaches including metabolomics and functional genomics.[5 25–27] In addition, several of these newer approaches have demonstrated potential for identifying individuals with probable chronic carriage. Identification and treatment of chronic carriers in a population would significantly improve our understanding of disease dynamics and allow targeted treatment strategies to reduce community transmission of infection.

Furthermore, while using these new diagnostic tests in comprehensive passive surveillance or even active surveillance programmes will identify acute cases of typhoid disease, the incidence rate of subclinical infection and exposure is unknown. These data are likely essential to understanding pathogen transmission as well as the development and maintenance of immunity against clinical disease[12] and are likely only to be obtainable through the use of seroepidemiological methods. Previous seroepidemiological studies have examined the cross-sectional age-stratified prevalence of antibodies to the *S* Typhi flagellar (H) antigen in Santiago, Chile,[28] and serum bactericidal antibody in Kathmandu, Nepal.[29] Also, high-antibody titres to the Vi (Vi capsular polysaccharide; virulence factor) antigen of *S* Typhi measured by agglutination assay have been used to estimate chronic carrier frequency in population-based studies,[30–32] although these studies were not universally successful at identifying

chronic carriers.[33] Improvements in serological assays and the discovery of a newer generation of diagnostic antigens[34] suggest that population-based serosurveillance studies, especially using longitudinal measurements, may provide key data regarding the incidence of subclinical infection with the bacterial agents of enteric fever and the prevalence of chronic carriers.

## Transmission

Closely related to the estimate of incidence is the contribution of different typhoid states to ongoing transmission within a community. Acute cases of enteric fever, individuals with subclinical infection, as well as chronic carriers likely all contribute to the transmission and maintenance of S Typhi and S Paratyphi A in endemic areas, but not to the same degree. This is an essential area of further research, particularly given the negative impact transmission from chronic carriers could have on the indirect protection afforded by typhoid vaccines.[12] The importance of 'short cycle' transmission via contaminated food and water in the immediate environment versus 'long cycle' transmission via the contamination of community water sources is also an area of direct relevance in the control of typhoid fever, primarily through non-vaccine public health interventions including water and sanitation hygiene (WASH) improvements.

## Bacterial genome variation

The enteric fever agents S Typhi and S Paratyphi A, while genetically distinct from other serovars of *Salmonella enterica*, exhibit low genetic diversity and whole-genome sequence analysis is required to uncover patterns of genetic relatedness between isolates.[35–37] These techniques have been used previously to track the development and spread of antibiotic resistance on a global scale[6] and have shown potential to accurately track transmission links between individuals, specifically within affected households, in addition to the broader community.[38] One use of this high-resolution typing previously has been to determine whether isolates obtained from those with acute infection are distinct from those found in chronic gall bladder carriers.[39]

## Host susceptibility

While environmental risk factors for enteric fever acquisition are thought to be relatively well understood, recent evidence has emerged implicating genetic factors in host susceptibility to infection.[35 36] Many early typhoid genetic susceptibility studies reporting candidate genes were limited by small sample sizes and choice of controls; a recent large-scale genome-wide association study. however, has identified a specific locus (HLA-DRB1*405) which confers fivefold protection against developing typhoid fever.[35] How genetic variation at this site influences protection against typhoid is still under investigation, but may result in functional differences in MHC class II amino acid sequences; in turn, this may influence S Typhi epitope selection and antigen presentation or the scale and format of the host's T cell response.

## Vaccines

Vaccines currently available for typhoid are either oral live-attenuated bacterial strains or are derived from the S Typhi polysaccharide capsule, called Vi; there are no licensed vaccines against S Paratyphi A or non-typhoidal salmonellae (NTS). Despite recommendations for their use, these vaccines have only sparsely been introduced in high-prevalence settings.[7 40] The reasons for this are unclear, but likely include failure of prioritisation and a short fall in advocacy, in addition to technical deficiencies of the vaccines themselves. One common reason cited is the T-cell-independent nature of the immune response to the Vi polysaccharide vaccine; this renders the vaccine of little use in young children below the age of 2 years. In addition, this feature means that the immune response does not boost, requiring repeat vaccination to be administered every 3–5 years. As with other thymus-independent vaccines, to overcome this problem the Vi capsule may be chemically conjugated to a protein carrier,[8] as has successfully been done with conjugation of Vi to *Pseudomonas aeruginosa* exotoxin A.[9] Despite demonstrating good immunogenicity and efficacy in field trials,[10 41] the vaccine is still unlicensed. Newer Vi-conjugate vaccines are in development and have progressed to assessment in field trials,[42 43] and the Oxford human typhoid challenge model.[44] Recent data from the challenge models suggest that the Vi response elicited by the vaccine is high and likely to be protective, at least in naive healthy adult volunteers with no immunity or those with history of infection.[44] Furthermore, a recent study in Indian school-children found a vaccine efficacy of 100% (95% CI 97.6% to 100%) after a single dose of Vi conjugated to tetanus toxoid during the first year of follow-up.[43]

The key issue remaining, once safe, well-tolerated, immunogenic vaccines become widely available, is how best to implement them in endemic settings. Currently available surveillance data from most regions are insufficient to demonstrate the age band with the highest incidence of enteric fever.[45] Designing vaccination strategies to cover the years during which children are most at risk and generating indirect protection by preventing infection among those age groups driving transmission could be facilitated through the use of well-informed mathematical modelling.

## THE STRATAA STUDY: RATIONALE AND AIMS

The Strategic Typhoid alliance across Africa and Asia (STRATAA) study draws together an international team of investigators, field sites, laboratories and research institutes to address many of the key outstanding questions described above regarding the burden of enteric fever and *Salmonella* exposure in endemic regions, the mechanisms of susceptibility and infection transmission. These have been identified as key uncertainties in

**Table 1** Overarching objectives of the Strategic Typhoid alliance across Africa and Asia study

| | |
|---|---|
| **Primary** | **To characterise the burden of enteric fever at three urban sites in Africa and Asia** |
| Secondary | Assess the burden/incidence of enteric fever |
| | Assess the seroincidence of infection |
| | Assess host factors affecting burden/incidence/transmission of enteric fever |
| | Assess effect of pathogen genetics on burden/incidence/transmission of enteric fever |
| | Develop diagnostic tools for rapid and consistent typhoid diagnosis |
| | Develop transmission modelling and modelling of vaccine introduction impact |
| Tertiary objectives | Strengthen research capacity in enteric fever endemic regions |
| | Provide data to appropriate institutes and governments to advocate vaccine implementation |
| | To characterise the burden of invasive non-typhoidal salmonellae and other invasive pathogens at an urban site in Malawi, Africa |

mathematical models.[11–13] By conducting field studies of the epidemiology and burden of enteric fever across three sites with distinct epidemiological profiles and applying state-of-the-art molecular methods, the goal of the STRATAA study is to collect the data needed to enhance our understanding of pathogen transmission, exposure and susceptibility. These data can then be used to rigorously parameterise and validate models for the transmission dynamics of the agents of enteric fever, such that these models can then be used to evaluate different vaccination strategies and, importantly, help predict the expected impact resulting from the direct and indirect effects of vaccine introduction. These and other over-arching objectives of the STRATAA study are listed in table 1.

To deliver on these objectives, the following studies will be performed in parallel at each of the three chosen field sites, starting in May 2016 with activities continuing until October 2018. First, in a well-defined population catchment of approximately 100 000 individuals, a detailed demographic census survey will be undertaken. Second, in healthcare facilities used by the census population (as confirmed through a healthcare utilisation survey) prospective passive surveillance to detect cases of enteric fever will be performed. Third, a seroincidence study will be conducted in an age-stratified sample of the census population at each site, collecting blood samples at intervals to estimate the rate of seroconversion to typhoidal *Salmonella* antigens and hence the rate of subclinical/asymptomatic exposure to these bacteria in the general population.

To meet the further objectives, which include evaluating diagnostic tests, identifying antimicrobial resistance patterns, exploring host susceptibility and bacterial virulence/genomic variation, biological samples, specimens and metadata will be collected during these individual studies. Data will be pooled to inform transmission dynamic and health economic models that can be used to help design future vaccine effectiveness studies and evaluate vaccine delivery strategies for widespread deployment.

## STUDY SITES

With known high incidence throughout South and Southeast Asia and a recent increase in reported cases from sub-Saharan Africa,[46 47] three sites were selected from across Africa and Asia based on known high rates of enteric fever and the research capacity to deliver a study of this size and logistical complexity. The three sites differ in their epidemiological profiles and history of enteric fever incidence.

### Dhaka, Bangladesh

Mirpur is an area located within Dhaka Metropolitan area, Bangladesh (figure 1A), situated 7 km northwest of the International Centre for Diarrhoeal Disease Research, Bangladesh (icddr,b) main campus. The icddr,b manages two hospitals in Dhaka city, one in the main campus in Dhaka (the main hospital) and another in the Mirpur area known as Mirpur Treatment Centre comprising a 50-bed inpatient facility. As part of this programme, these and other health facilities (≥10 total) will be kept under careful surveillance for enteric fever in the census area. The total catchment area for the field site for the STRATAA study is $10.79\,km^2$ with a total population of about 603 658 at a density of $55\,946/km^2$. Approximately 98% of residents have access to tap water supplied by the municipality while the remainder use wells, hand pumps and other sources such as ponds and rivers in the study area.

Previous data from blood culture surveillance revealed a high incidence of disease within Dhaka, with the burden particularly high in children under 5 years of age (estimated at 18.7 episodes/1000 person-years in this age group). The current *S* Typhi:Paratyphi ratio is approximately 5:1. Typhoid fever is reported throughout the year but peaks during the monsoon season.[48 49] Of note, 15% of *S* Typhi isolates are multi-drug resistant (MDR) and around 97% isolates are resistant to nalidixic acid[50]; extended-spectrum beta-lactamase-producing organisms have also been isolated from enteric fever patients in this setting.

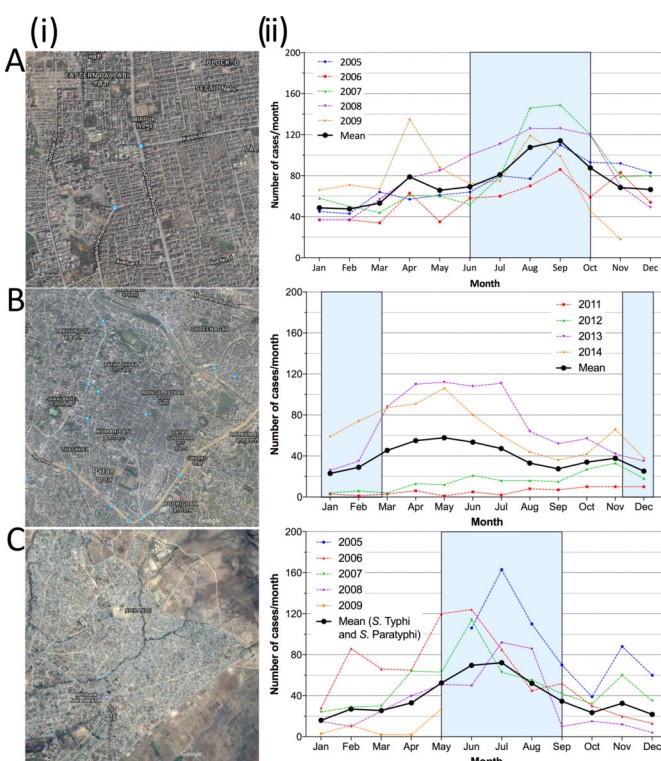

**Figure 1** Description of the Strategic Typhoid alliance across Africa and Asia study field sites, demonstrating (i) the location of the three sampling sites in (A) Mirpur (Dhaka, Bangladesh), (B) Ndirande (Blantyre, Malawi) and (C) Patan (Kathmandu, Nepal); (ii) the historical number of typhoid cases detected per month at each site (blue box marks the annual monsoon season). S Typhi, *Salmonella enterica* serovar Typhi; S Paratyphi A, *Salmonella enterica* serovar Paratyphi A.

## Blantyre, Malawi

Ndirande is a large urban township on the outskirts of Blantyre city, Malawi, 6 km from the main referral hospital (figure 1B). It has a young population of around 100 000 people spread over 6.77 km². It is serviced by one health clinic staffed by clinical officers. It has high reported rates of typhoid fever along with a population HIV prevalence of around 18%.[51]Queen Elizabeth Central Hospital (QECH) in Blantyre, Malawi, is the government-funded hospital for Blantyre district, serving a local population of 1.3 million persons and provides tertiary care to southern Malawi.

NTS have previously been the most common cause of invasive bloodstream infections in Blantyre,[52] but since 2011 there has been a rapid increase in the number of S Typhi cases seen at QECH, from approximately 14/year between 1998 and 2010 to 782 in 2014.[46] Much of this increase has been due to the emergence of the H58 clone and is associated with an MDR phenotype.[47 53] This outbreak appears unrelated to HIV infection, but has resulted in high rates of mortality (2.5% adult and paediatric) despite the availability of fluoroquinolone antibiotics.[47] In this setting, enteric fever is seasonal, with the peak number of cases seen at the end of the wet season and during the early dry season when the prevalence of malnutrition is also highest.

## Patan, Nepal

Patan is located within the Lalitpur Sub-Metropolitan City within the Kathmandu Valley, Nepal (figure 1C). The population is generally poor, with most living in overcrowded conditions and obtaining their water from stone spouts or sunken wells. Patan Hospital is a 318-bed government hospital providing emergency and elective outpatient and inpatient services to this area. The local catchment population of the hospital is approximately 200 000 people in about 20 km², with a population density of 8000/km²; there is a high rate of immigration for employment, particularly young males from rural areas. Enteric fever is frequently managed in the outpatient clinic at Patan Hospital, which has approximately 200 000 outpatient visits annually.

Uptake of typhoid Vi vaccination is limited and natural exposure/subclinical infection is common.[29] Approximately 400 culture-confirmed cases of enteric fever are diagnosed at Patan Hospital each year, with a peak during the monsoon months. The current S Typhi:Paratyphi A ratio is approximately 1:1.[54] Antimicrobial resistance is more commonly observed in S Paratyphi A isolates; however, emergence of fluoroquinolone-resistant S Typhi isolates has also been recently identified,[55] whereas MDR strains of either serovar are rare.[38]

## METHODS

### Population census

In order to accurately calculate an incidence rate of typhoid fever for each of the study sites, a demographic census will be performed at baseline and repeated at 2 years. The objective of the census is to identify/characterise the source population corresponding to the catchment areas for the passive surveillance sites described below and estimate the person-time under surveillance.

Census data will be updated with births, deaths and migrations every 6 months in Dhaka and Patan, and at 2 years in Blantyre. The number of participants for the demographic census survey in each site needed to produce a two-sided 95% CI with a precision (half-width) of 50% for the anticipated typhoid incidence rate detected through passive clinical surveillance has been calculated (table 2). The necessary catchment population size is driven by the size required to estimate the expected incidence rate in the 0–4-year-old age group, which is estimated to be approximately 10% of the total population in each site.

The census will take place within a demarcated geographic area that is a known catchment population for the surveillance sites (figure 1). In total, at least 100 000 individuals will be enumerated from ≥20 000 households. The head/key informant within the household will provide written informed consent to take part in the study. Information on all residents within the household

**Table 2** Sample size required for the target populations in the three sites to estimate annual, blood culture-confirmed typhoid incidence in passive clinical surveillance

| Age groups (years) | Anticipated typhoid incidence per 1000 persons* | Precision or half-width | Sample size required |
|---|---|---|---|
| 0–4 | 1.5 | 0.75 | 10 125 |
| 5–14 | 1.0 | 0.5 | 15 119 |
| >14 | 0.5 | 0.25 | 29 801 |

*Assumed age-specific incidence rates (based on data from Dhaka, Bangladesh, Delhi, India, and Dong Thap, Vietnam).

at the time of the census will be gathered from the head of the household/key informant.

A household is defined as individuals living in the same dwelling or compound and sharing food from the same kitchen. A household member is considered to have migrated out if she/he has left the household and does not intend to come back within 6 months of the time she/he left. A person is considered to have migrated in if she/he was not previously included in the list of household members and intends to live in the household for the next 6 months. At enrolment, the head of each household will be made aware of the passive surveillance component of the study and encouraged to use the field-site facilities capturing acute febrile illnesses, including acute enteric fever cases. The characterised census population will form the sampling frame for the further components (passive surveillance, healthcare utilisation/WASH surveys and serosurvey) described below.

## Passive surveillance

The passive surveillance component of STRATAA is designed to capture cases of febrile disease occurring in each of the three census populations. Patients presenting to any of the clinical surveillance sites with a history of subjective fever >72 hours or objective fever >38.5°C on presentation will be approached for enrolment.

An index case will be defined as an individual with a blood culture result confirming infection with S Typhi or S Paratyphi A (or NTS in Malawi), and whose household is included in the census survey. These cases will be used

to calculate the disease incidence in each of the three census sites. Consenting individuals will have samples of blood, urine and stool collected to determine a diagnosis of S Typhi, S Paratyphi A or NTS and to provide material for the further diagnostic and genetic aims of the study (see online supplementary table 1). Patients not resident in the census area will be enrolled as additional cases for the laboratory and genetic components of the programme if written consent is provided.

## Healthcare utilisation and water, sanitation and hygiene surveys

To characterise the healthcare-seeking behaviour of individuals living in the census areas, at least 735 households will be randomly selected from the census area at each site to participate in a healthcare utilisation survey. Data will be collected from the head of the household/key informant to describe the actual and hypothetical usage of healthcare facilities for febrile episodes. The aim of this component is to estimate the per cent of cases (fulfilling the fever case definition) in each age stratum (0–4 years, 5–14 years and >14 years of age) who would or would not seek attention at one of the designated passive surveillance health facilities. Additional data regarding sanitation and hygiene facilities and usage will also be collected. Annual data collection periods will coincide with the peak typhoid season at each of the three sites.

## Serosurveys

To estimate the seroincidence and seroprevalence of clinical (both formally diagnosed and undiagnosed cases) and subclinical infection or exposure to S Typhi and S Paratyphi, systematic serosurveillance will be performed at each site. Blood samples will be collected at baseline and 3 months later, with sample collection initiated in an ongoing basis over the course of 1 year (table 3). To identify participants for the serosurvey, an age-stratified approach will be used to randomly select individuals from the census population from each age group. Suitable participants will be enrolled by field workers; where the individual identified is not available, a household member in the same age group will be selected. Where this is not possible, further households will be randomised into this

**Table 3** Sample size calculations for the serological surveys to estimate age-specific rates of high titres of serum IgG anti-H(d) antibodies

| | Anticipated seroincidence (%) | Initial samples (n) | Anticipated number of events | Follow-up samples (n)* | Events detected (n) | Binomial exact (95% CI) | Probability of observing 0 events |
|---|---|---|---|---|---|---|---|
| 0–4 years | 0.2 | 2500 | 5 | 2000 | 4 | 0.0005 to 0.0051 | 0.0182 |
| 5–9 years | 0.4 | 1300 | 5.2 | 1040 | 4 | 0.0010 to 0.0098 | 0.0155 |
| 10–14 years | 0.8 | 800 | 6.4 | 640 | 5 | 0.0025 to 0.0181 | 0.0059 |
| >14 years | 0.2 | 3900 | 7.8 | 3120 | 6 | 0.0007 to 0.0042 | 0.0019 |
| Total | | 8500 | 24.2 | 6800 | 19 | | |

Assumes age-stratified individual sampling; also accounts for detection of 1% chronic carriage rate in those aged ≥10 years.
*Assuming 5% migration and 15% refusal.

component to ensure adequate numbers of individuals in the different age groups.

Seroincidence (indicative of recent infection) will be calculated by measuring the rate of seroconversion to anti-H(d) (anti-flagellin) IgG and other acute-phase antibodies between both time points, with the denominator consisting only of individuals sampled twice and seronegative at the first time point.[56]

The seroprevalence of anti-Vi IgG will also be measured at baseline to identify individuals who could be potential chronic carriers. Previous studies have used high anti-Vi IgG antibody levels in serum as a marker of chronic carriage, and our aim is to use a recently validated/approved anti-Vi ELISA method to determine possible rates of chronic carriage across the three populations.[32 57 58] In order to validate this method and to identify possible chronic carriers in the population (in addition to identifying whether there may be specific host genetic risk factors for chronic carriage), those individuals with a 'high' serum anti-Vi IgG titre will be reapproached and asked to provide two stool samples within 48 hours. With the agreement of relevant local ethics committees, identified chronic carriers (confirmed high anti-Vi IgG serum sample with a positive stool culture) will be treated with antibiotics for chronic carriage and clearance of bacterial shedding will be confirmed.

## Household contacts

To further investigate possible transmission links, household contacts of index cases presenting with blood culture-confirmed *S* Typhi or *S* Paratyphi A infection will be identified through the census data collection and approached to take part in this component of the study. Up to five members of the household, with consent, will be asked to provide a blood and stool sample at the time of discharge of the index case, a further stool sample at 1 month and repeat serology at 6 months (see online supplementary table 1), and asked about the occurrence of symptoms. These samples will be investigated to identify those with subclinical infection or possible chronic carriers. Secondary attack rates for the occurrence of symptomatic infection will be estimated. If serology suggests chronic carriage in a household contact, additional blood and stool samples will be collected to confirm this, and antimicrobial treatment to eradicate carriage will be proposed.

Those individuals who are identified as shedding bacteria but without high serum anti-Vi IgG levels will be followed up at a 1-year interval to repeat stool culture. Repeat blood samples will be collected from household contacts approximately 3 months after initial sampling to explore whether rates of seroconversion are higher in these individuals compared with those in the general population. At least 73 households of index cases are expected to be enrolled in this component of the study.

In a similar approach to the acute cases of typhoid infection, where chronic carriers are identified from the serosurveys, household transmission studies will be performed among household contacts of possible chronic carriers (ie, those with a 'high' serum anti-Vi IgG titre). A one-time attempt will be made to enrol up to five members of the household, sampling serology and stool looking for evidence of typhoid infection. This will provide data on secondary attack rates for both acute and chronic typhoid states.

## Data management and analysis

Census and serosurvey data collection forms will be developed through a structured iterative process and then implemented using Open Data Kit,[59] a system enabling electronic mobile data collection, with customisations by Nafundi, USA, on Android-based tablets. Each household within the census will be assigned a unique study ID and geo-located using GPS where possible; individuals will be given a member number within the household. This information will be collected by local enumerators over a 1–4-month period via these forms and adapted to the three geographic settings. Data will be uploaded onto MySQL databases, where SQL routines will be run nightly to enforce data cleaning on critical variables beyond ODK's validation routines. Daily anonymised data will be backed up from the three sites centrally. For the passive surveillance and household contacts studies, a combination of tablet and paper-based case report forms will be used to capture the data. Data from paper forms will be transcribed onto electronic databases using Open Clinica.[60] Database reports and descriptive analyses will be generated weekly. To assess efficiency and quality of data capture, the volume, accuracy and time of data collection can be quantified.

The distribution and burden of enteric illness is likely to vary between countries and thus analyses will be conducted separately for each country. Where data are combined across countries, an adjustment for country differences will be included in statistical models.

## ETHICS

Written informed consent was obtained from the head of each household (as the 'key informant') on behalf of the entire household in the demographic census and healthcare utilisation surveys. In each of the other components, individual written informed consent was obtained from individuals over the age of 18 or by a parent/guardian from individuals below this age with additional assent sought from those aged between 11 and 17 years.

This protocol has received ethics approval from the Oxford Tropical Research Ethics Committee, the Malawian National Health Sciences Research Committee and University of Malawi Research Ethics Committee, the Nepal Health Research Council and the icddr, b Institutional Review Board.

## DISSEMINATION

We hope to make the results from these studies widely available and plan to disseminate our analyses in

international peer-reviewed journals. Investigators will be involved in reviewing drafts of the manuscripts, abstracts, press releases and any other publications arising from the study. Furthermore, data from these studies will also be used in the submission of postdoctoral theses.

## COMMUNITY PUBLIC ENGAGEMENT

A collection of specific activities engaging the local population with both the subject of typhoid fever and the activities of the study have been carried out. In Malawi, for example, community leaders have been informed and consulted on certain aspects of the study, shown the proposed activities of study teams and given tours of research facilities. There has been engagement with various forms of media disseminating information on the importance of typhoid and study aims. In Nepal, field staff have been given detailed information to communicate to the local populations.

Further activities are planned to ensure the local populations are informed and engaged.

## DISCUSSION

The STRATAA study is a comprehensive multicentre study aiming to improve understanding of typhoidal *Salmonella* infection in high-risk endemic populations. This study has been designed to answer key questions and data gaps identified through an innovative application of recent mathematical modelling. These gaps include measuring the burden of age-stratified disease, identifying the relative contribution of asymptomatic/subclinical *Salmonella* infection/exposure and estimating the contribution to ongoing transmission from the chronic carrier state. These data will inform further modelling required to develop and optimise disease control and prevention strategies that will eventually lead to disease elimination.

### Author affiliations
[1]The Hospital for Tropical Diseases, Wellcome Trust Major Overseas Programme, Oxford University Clinical Research Unit, Ho Chi Minh City, Vietnam
[2]Department of Paediatrics, Oxford Vaccine Group,University of Oxford, and the NIHR Oxford Biomedical Research Centre, Oxford, UK
[3]Department of Infection, Immunity and Cardiovascular Disease, University of Sheffield Medical School, Sheffield, UK
[4]International Centre for Diarrhoeal Diseases Research, Dhaka, Bangladesh
[5]Oxford University Clinical Research Unit, Patan Academy of Health Sciences, Kathmandu, Nepal
[6]Malawi Liverpool Wellcome Trust Clinical Research Programme, University of Malawi College of Medicine, Blantyre, Malawi
[7]Nuffield Department of Medicine, Centre for Tropical Medicine and Global Health, University of Oxford, Oxford, UK
[8]The Wellcome Trust Sanger Institute, Cambridgeshire, UK
[9]UCLA Fielding School of Public Health, Los Angeles, USA
[10]Mahidol-Oxford Tropical Medicine Research Unit,Mahidol University, Bangkok, Thailand
[11]The Peter Doherty Institute for Infection and Immunity,The University of Melbourne, Melbourne, Australia
[12]Institute of Infection and Global Health,University of Liverpool, Liverpool, UK
[13]Division of Infection and Immunity, University College London, London, UK
[14]Centre for Systems Genomics,University of Melbourne, Victoria, Australia
[15]Department of Biochemistry and Molecular Biology, University of Melbourne, Victoria, Australia
[16]Department of Epidemiology of Microbial Diseases, Yale School of Public Health, Yale University, New Haven, Connecticut, USA

**Collaborators** Abhilasha Karkey, Sabina Dongol, Amit Aryjal (Oxford University Clinical Research Unit, Patan Academy of Health Sciences, Kathmandu, Nepal); Nirod Chandra Saha (International Center for Diarrhoeal Diseases Research, Dhaka, Bangladesh); Tikhala Makhaza Jere, Chisomo Msefula, Tonney Nyirenda (University of Liverpool, UK, and the Malawi Liverpool Wellcome Trust Clinical Research Programme, Malawi); Tan Trinh Van (The Hospital for Tropical Diseases, Wellcome Trust Major Overseas Programme, Oxford University Clinical Research Unit, Ho Chi Minh City, Vietnam); Stephen Reece (Wellcome Trust Sanger Institute, Cambridge, UK); Merryn Voysey, Christoph J. Blohmke, Yama Farooq, Jennifer Hill (Oxford Vaccine Group, Department of Paediatrics, University of Oxford, and the NIHR Oxford Biomedical Research Centre, Oxford, UK); Neil J. Saad (Yale School of Public Health, Yale University, New Haven, Connecticut, USA).

**Contributors** SB, BB, JDC, GD, CD, MAG, RSH, VEP, FQ, KZ, SD, KH and AJP contributed to the conception and design of the study. ST drafted the protocol of the study. This manuscript was drafted by TD and JM. SB, BB, JDC, GD, SD, CD, MAG, KH, RSH, VEP, FQ, ST, KZ, MAK, FK, MS, DT and AJP read and critically revised the protocol and this manuscript prior to submission.

**Funding** Funding for the STRATAA study has been provided by a Wellcome Trust Strategic Award (no. 106158/Z/14/Z) and the Bill and Melinda Gates Foundation (no. 617 OPP1141321). The Malawi-Liverpool-Wellcome Programme and the Oxford University Clinical Research Unit in Vietnam are supported by the Wellcome Trust with Major Overseas Programme core awards.

**Disclaimer** Neither funding body had any role in designing the study, writing this manuscript or the decision to submit.

**Competing interests** AJP chairs the UK Department of Health's (DH) Joint Committee on Vaccination an Immunisation (JCVI) and the European Medicines Agency (EMA) Scientific Advisory Group on Vaccines and is a member of WHO's SAGE. The views expressed in this manuscript do not necessarily reflect those of JCVI, DH, EMA or WHO. AJP has previously conducted clinical trials on behalf of the University of Oxford funded by vaccine manufacturers but has no personal financial interests.

**Ethics approval** Oxford Tropical Research Ethics Committee, icddr, b Institutional Review Board, Nepal Health Research Council, Malawi National Health Sciences Research Committee.

**Provenance and peer review** Not commissioned; externally peer reviewed.

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
