## [Reviewer comments · BMJ Open]

ARTICLE DETAILS

TITLE (PROVISIONAL)	THE STRATAA STUDY PROTOCOL: A PROGRAMME TO ASSESS THE BURDEN OF ENTERIC FEVER IN BANGLADESH, MALAWI AND NEPAL USING PROSPECTIVE POPULATION CENSUS, PASSIVE SURVEILLANCE, SEROLOGICAL STUDIES AND HEALTHCARE UTILISATION SURVEYS
AUTHORS	Darton, Thomas; Meiring, James; Tonks, Susan; Khan, Md Arifuzzaman; Khanam, Farhana; Shakya, Mila; Thindwa, Deus; Baker, Stephen; Basnyat, Buddha; Clemens, John; Dougan, Gordon; Dolecek, Christiane; Dunstan, Sarah; Gordon, Melita; Heyderman, Robert; Holt, Kathryn; Pitzer, Virginia E.; Qadri, Firdausi; Zaman, K; Pollard, Andrew

VERSION 1 - REVIEW

REVIEWER	Carlos Franco Hospital Infantil de Mexico, Federico Gomez, and Phoebe Putney Memorial Hospital, Albany GA.
REVIEW RETURNED	12-Feb-2017

GENERAL COMMENTS	This is an important contribution.
------------------------------------

REVIEWER	Dr. Florian Marks International Vaccine Institute, Seoul, Korea In informative collaboration with the authors to streamline some activities between this protocol and the Severe Typhoid in Africa (SETA) program, which is being led by the reviewer. Collaborative working relationship with Dr. Stephen Baker.
REVIEW RETURNED	05-Mar-2017

GENERAL COMMENTS	This protocol is well written and addresses the research questions that are investigated.
---

REVIEWER	Steve Luby Stanford University United States
REVIEW RETURNED	01-May-2017

GENERAL COMMENTS	A clear description of an important study that will improve understanding of transmission and modeling estimates in these dense urban populations. Because typhoid transmission and incidence is characteristically heterogeneous, the tiny geographies
---

	purposely selected for their high incidence will be less informative of national or global burden of disease. I have only minor comments Minor comments 1. The manuscript should clarify how the health care utilization questions will be asked. Given the sample size of 750 per site, this reviewer assumes that respondents will be asked where they would go if a member of the household had a syndrome consistent with typhoid fever. A rich literature in psychology and behavioral economics suggests that people's responses to hypothetical scenarios is a poor predictor of their actual behavior. The authors may want to review this literature and consider the implication on their incidence estimates and design. 2. What will the study team do if the person targeted for the sero survey is not available, nor is there anyone else in the household within the targeted age group? 3. How many stool specimens will be collected to look for shedding of Salmonella typhi among suspect carriers? 4. Line 537: Replace 'AA' with 'A'
--	---

VERSION 1 – AUTHOR RESPONSE

Many thanks for the comments of all reviewers.

To respond to Professor Luby's comments;

1. The study is seeking to primarily collect data on actual healthcare seeking behavior. Where this is absent, hypothetical questions will be asked as described on line 514. A review of the literature on this aspect would be beneficial and will be suggested to the consortium.
2. If the participant selected for the serosurvey is absent, and no replacement in the household can be found further individuals have been randomized for each site from the demographic census to ensure that the recruitment targets are met for each age group. This extra detail has been added to the manuscript from line 532.
3. This detail is included on line 556 of the manuscript. The study team will aim to collect two stool samples within 48 hours from participants with high Vi titres that are suspected of chronic carriage.
4. On line 537 the 'AA' has been changed to 'A'.

Many thanks again for taking the time to consider our submission and for your comments.

VERSION 2 – REVIEW

REVIEWER	Stephen Luby Stanford University USA
REVIEW RETURNED	21-May-2017

GENERAL COMMENTS	The authors have addressed my concerns.
---